# Attitudes and Difficulties Associated with Benzodiazepine Discontinuation

**DOI:** 10.3390/ijerph192315990

**Published:** 2022-11-30

**Authors:** Masahiro Takeshima, Yumi Aoki, Kenya Ie, Eiichi Katsumoto, Eichi Tsuru, Takashi Tsuboi, Ken Inada, Morito Kise, Koichiro Watanabe, Kazuo Mishima, Yoshikazu Takaesu

**Affiliations:** 1Department of Neuropsychiatry, Akita University Graduate School of Medicine, 1-1-1 Hondo, Akita City 010-8543, Japan; 2Psychiatric and Mental Health Nursing, St. Luke’s International University, 10-1 Akashi-cho, Chuo-ku, Tokyo 104-0044, Japan; 3Department of Neuropsychiatry, Kyorin University School of Medicine, 6-20-2 Shinkawa, Mitaka, Tokyo 181-8611, Japan; 4Department of General Internal Medicine, St. Marianna University School of Medicine, 1-30-37 Shukugawara, Kawasaki 214-8525, Japan; 5Katsumoto Mental Clinic, 10-13 Horikoshicho, Tennoji-ku, Osaka City 543-0056, Japan; 6Department of Neurosurgery, Munakata Suikokai General Hospital, 5-7-1 Himakino, Fukutsu-shi 811-3298, Japan; 7Department of Psychiatry, School of Medicine, Kitasato University, Kitazato, Sagamihara shi 252-0329, Japan; 8Centre for Family Medicine Development, Japanese Health and Welfare Co-Operative Federation, 3-25-1 Hyakunincho, Shinjuku-ku, Tokyo 169-0073, Japan; 9Department of Neuropsychiatry, Graduate School of Medicine, University of the Ryukyus, 207 Aza-Uehara, Nishihara-cho, Nakagami-gun, Nishihara 903-0215, Japan

**Keywords:** benzodiazepine, cognitive behavioral therapy, cross-sectional survey, discontinuation, psychiatrist, switching

## Abstract

Long-term use of benzodiazepine receptor agonists (BZDs) may depend on clinicians’ BZD discontinuation strategies. We aimed to explore differences in strategies and difficulties with BZD discontinuation between psychiatrists and non-psychiatrists and to identify factors related to difficulties with BZD discontinuation. Japanese physicians affiliated with the Japan Primary Care Association, All Japan Hospital Association, and Japanese Association of Neuro-Psychiatric Clinics were surveyed on the following items: age group, specialty (psychiatric or otherwise), preferred time to start BZD reduction after improvement in symptoms, methods used to discontinue, difficulties regarding BZD discontinuation, and reasons for the difficulties. We obtained 962 responses from physicians (390 from non-psychiatrists and 572 from psychiatrists), of which 94.0% reported difficulty discontinuing BZDs. Non-psychiatrists had more difficulty with BZD discontinuation strategies, while psychiatrists had more difficulty with symptom recurrence/relapse and withdrawal symptoms. Psychiatrists used more candidate strategies in BZD reduction than non-psychiatrists but initiated BZD discontinuation after symptom improvement. Logistic regression analysis showed that psychosocial therapy was associated with less difficulty in BZD discontinuation (odds ratio, 0.438; 95% confidence interval, 0.204–0.942; *p* = 0.035). Educating physicians about psychosocial therapy may alleviate physicians’ difficulty in discontinuing BZDs and reduce long-term BZD prescriptions.

## 1. Introduction

Benzodiazepine receptor agonists (BZDs) are widely used psychotropic medications [1] that increase GABAergic neural activity by acting on the benzodiazepine receptor to induce sedative, hypnotic, anxiolytic, anticonvulsant, amnesic, and relaxing effects [2]. Short-term BZD treatment is recommended for chronic insomnia and anxiety disorders because of its immediate effects and lack of serious side effects [3,4,5,6,7,8,9,10]. Although long-term BZD treatment is not recommended because of its unclear long-term beneficial effects and published adverse effects, such as dependence [11], falls [12], and fractures [13], long-term prescription of BZDs continues to be practiced and is an international public health issue [14,15,16,17].

One possible reason for long-term BZD prescriptions may be inadequate awareness of BZD discontinuation strategies and difficulties with BZD discontinuation among physicians. Despite repeated warnings from governments and academic societies against long-term BZD use [18,19,20], specific BZD discontinuation strategies (i.e., when to start discontinuing BZDs after symptom improvement, how to discontinue BZDs, and how much the patient should be stabilized before discontinuation of BZDs) are quite limited. We proposed the following hypotheses based on previous studies: (1) Physicians who start BZD reduction later are more likely to fail to discontinue BZDs and may face more difficulty than those who do not because of long-term use of BZDs increases the risk of physical dependence and withdrawal symptoms [21]. (2) Physicians who use abrupt discontinuation are more likely to fail to discontinue BZDs and face more difficulty discontinuing BZDs than those who use the gradual tapering method because abrupt discontinuation may induce withdrawal symptoms and rebound insomnia and anxiety [22,23,24]. Cognitive behavioral therapy [25,26,27], switching to other psychotropic medications (e.g., paroxetine, valproate, and carbamazepine) [28,29], and brief interventions (e.g., general physician’s (GP’s) letter, booklets) [30] have been reported to assist in BZD discontinuation, albeit without robust evidence. Physicians who do not use them are more likely to fail and have more difficulty discontinuing BZD than those who do. Evaluation of these hypotheses may provide clues to solving the problem of long-term BZD use.

Previous surveys examining attitudes toward BZD prescriptions have focused on GPs, not psychiatrists [31,32,33]. Since psychiatrists have more opportunities to treat patients with anxiety and insomnia symptoms and perform cognitive behavioral therapy, potentially facilitating BZD discontinuation in patients with psychiatric disorders [34], comparisons of BZD discontinuation strategies and difficulties between psychiatrists and non-psychiatrists and investigation of the factors related to difficulties in BZD discontinuation may be worthwhile.

Therefore, we conducted a questionnaire survey to investigate the differences in the strategies and difficulties in BZD discontinuation between psychiatrists and non-psychiatrists and to identify factors related to difficulties in BZD discontinuation.

## 2. Materials and Methods

### 2.1. Participants and Procedure

This study used a cross-sectional questionnaire survey. We sent questionnaires between 22 October 2021 and 1 February 2022 to all physicians affiliated with the Japan Primary Care Association (JPCA), the All Japan Hospital Association (AJHA), and the Japanese Association of Neuro-Psychiatric Clinics (JAPC) through the three associations; JPCA and AJHA were contacted by e-mail and JAPC by letters. The JPCA members joined the association in agreement with its purpose of conducting continuous and comprehensive health, medical, and welfare practices and academic activities that value the connection with residents so that people can lead healthy lives. The AJHA members were representatives of hospitals who joined the association in agreement with its purpose of contributing to the improvement of public health and the development of local communities by conducting surveys, research, and other activities necessary for the improvement and development of hospitals and fulfillment of their missions. The JAPC members were managing physicians or equivalent physicians of clinics that advocate for psychiatry and engaged in medical treatment with psychiatry as the focus of their practice. Japan does not have a GP system, and physicians can prescribe hypnotics or anxiolytics without board certification in psychiatry or sleep medicine. In Japan, among patients prescribed hypnotics for the first time, approximately 94% were prescribed hypnotics by a nonpsychiatric department [35].

### 2.2. Survey Items

The questionnaire used in this study consisted of the following items:Age (20 s, 30 s, 40 s, 50 s, 60 s, 70 s, and 80 s and over).Specialty (psychiatry or otherwise).Preferred timing to start BZD reduction after symptom improvement (immediately, within 3 months, >3 months but <6 months, >6 months but <12 months, after 12 months, no need to reduce BZDs if there are no side effects, I do not know, or others).Methods used to discontinue BZD anxiolytics:✓Gradual tapering (yes or no);✓Abrupt discontinuation (yes or no);✓Switching to other psychotropic drugs (long-acting benzodiazepine anxiolytics or psychotropic drugs with anxiolytic effects, such as selective serotonin reuptake inhibitors (SSRIs) and tandospirone) (yes or no);✓Psychosocial therapy, patient materials, and pamphlets (yes or no).Methods used to discontinue BZD hypnotics:✓Gradual tapering (yes or no);✓Abrupt discontinuation (yes or no);✓Switching to other psychotropic drugs (novel hypnotics such as melatonin receptor agonists or orexin receptor antagonists, or psychotropic drugs with sedative effects such as trazodone and quetiapine) (yes or no);✓Psychosocial therapy or patient materials and pamphlets (yes or no).Difficulties in discontinuation of BZDs (present or absent).Reasons for the difficulties in discontinuation of BZDs:✓I do not know how to discontinue (yes or no);✓I do not know when to discontinue (yes or no);✓I do not know the extent to which the patient should be stabilized before discontinuing BZDs (yes or no);✓Attempted to discontinue treatment but could not because of recurrence/relapse of symptoms (yes or no);✓Attempted to discontinue but could not because of withdrawal symptoms (yes or no);✓Attempted to discontinue but could not because of the patient’s resistance (yes or no).

In this study, if the same reduction method was used for both hypnotics and anxiolytics, we considered that the method was used to discontinue BZDs.

### 2.3. Statistical Analysis

Categorical variables are expressed as numbers (%). Chi-square and Z tests were performed to examine the differences between the psychiatrist and non-psychiatrist groups. A binary logistic regression analysis was performed to examine factors associated with difficulties in BZD discontinuation by considering age group, specialty, and preferred timing to start BZD reduction after symptoms improved. Statistical significance was set at *p* < 0.05 (two-sided). All statistical analyses were performed using SPSS Statistics 28.0 (IBM Corp. Armonk, NY, USA).

### 2.4. Ethics

The ethics committee of St. Luke’s International University (2021-604) approved this study. Informed consent was obtained from the participants in written or electronic form before answering the questionnaire.

## 3. Results

### 3.1. Demographics

In this survey, the response rate was 4.73% (251/5306), 6.62% (168/2537), 32.1% (543/1690), and 10.1% (962/9533) at JPCA, AJHA, JAPC, and overall, respectively. Among the 962 participants, 26.1% belonged to JPCA, 17.5% to AJHA, and 56.4% to JAPC. Evaluation of the medical specialty of the participants showed that 40.5% (390/962) were non-psychiatrists, and 59.5% (572/962) were psychiatrists. There were no residents or medical students among the participants.

### 3.2. Difference between Psychiatrists and Non-Psychiatrists in This Survey (Table 1)

#### 3.2.1. Age Groups

Most participants were middle-aged or older, with 29.5% in their 60 s, 28.3% in their 50 s, and 18.7% in their 40 s. Compared with psychiatrists, non-psychiatrists were more likely to be in their 20 s, 30 s, and 40 s, while fewer were in their 60 s and 70 s.

#### 3.2.2. Preferred Timing to Start BZD Reduction after Improvement in Symptoms

Among the 962 participants, 19.4%, 35.6%, 20.2%, 7.1%, and 1.9% preferred to start BZD reduction immediately after improvement, within 3 months after improvement, within 6 months after improvement, within 12 months after improvement, and 12 months after improvement; 6.0% answered that benzodiazepine reduction is not necessary if there are no side effects; and 2.8% answered, “I don’t know”. Non-psychiatrists, compared with psychiatrists, were likely to start BZD reduction immediately after symptom improvement, and fewer non-psychiatrists started BZD reduction within six or 12 months after symptom improvement.

#### 3.2.3. Methods Used for Discontinuation of BZDs

Gradual tapering was used by 83.8% of the participants for BZD anxiolytics, 81.7% for BZD hypnotics, and 77.1% for both; abrupt discontinuation was used by 3.5% of participants for BZD anxiolytics, 3.1% for BZD hypnotics, and 2.0% for both; switching was used by 48.1% of participants for BZD anxiolytics, 62.8% for BZD hypnotics, and 41.4% for both; psychosocial therapy was used by 19.1% of participants for BZD anxiolytics, 18.4% for BZD hypnotics, and 14.6% for both; and patient materials and pamphlets were used by 5.1% of participants for BZD anxiolytics, 6.5% for BZD hypnotics, and 4.3% for both. Non-psychiatrists, compared with psychiatrists, were more likely to use abrupt discontinuation (*p* = 0.003) and less likely to use gradual tapering, switching, and psychosocial therapy (*p* < 0.001, *p* < 0.001, and *p* < 0.001, respectively). The use of patient materials and pamphlets for discontinuation of BZDs did not differ between non-psychiatrists and psychiatrists.

#### 3.2.4. Difficulties with BZD Discontinuation and Their Details

Most physicians experienced difficulties in BZD discontinuation, with no difference between psychiatrists and non-psychiatrists. Among the reported difficulties in BZD discontinuation, patient resistance was the most common difficulty reported by both non-psychiatrists and psychiatrists, followed by relapse or recurrence of symptoms. In comparison with psychiatrists, non-psychiatrists experienced more difficulty in discontinuing BZDs and the conditions in which they could be discontinued (*p* < 0.001, *p* < 0.001, and *p* < 0.001, respectively) and less difficulty with relapse/recurrence of symptoms and withdrawal symptoms (*p* < 0.001 and *p* < 0.001, respectively).

**Table 1 ijerph-19-15990-t001:** Demographics of the respondents.

	Total (*N* = 962)	Non-Psychiatrists (*N* = 390)	Psychiatrists (*N* = 572)		
	Number (%)	Number (%)	Number (%)	*p*-Value	Post-hoc
Age group				<0.001 *	
20 s	12 (1.2%)	11 (2.8%)	1 (0.2%)		NP > P
30 s	85 (8.8%)	74 (19.0%))	11 (1.9%)		NP > P
40 s	180 (18.7%)	102 (26.2%)	78 (13.6%)		NP > P
50 s	272 (28.3%)	109 (27.9%)	163 (28.5%)		NS
60 s	284 (29.5%)	79 (20.3%)	205 (35.8%)		NP < P
70 s	109 (11.3%)	13 (3.3%)	96 (16.8%)		NP < P
80 s and more	18 (1.9%)	2 (0.5%)	16 (2.8%)		NS
Non-response	2 (0.2%)	0 (0%)	2 (0.3%)		NS
Preferable timing to start BZDs reduction after symptoms improve				<0.001 *	
Immediately	187 (19.4%)	104 (26.7%)	83 (14.5%)		NP > P
Within 3 months	342 (35.6%)	128 (32.8%)	214 (37.4%)		NS
More than 3 months but less than 6 months	194 (20.2%)	64 (16.4%)	130 (22.7%)		NP < P
More than 6 months but less than 12 months	68 (7.1%)	19 (4.9%)	49 (8.6%)		NP < P
After 12 months	18 (1.9%)	4 (1.0%)	14 (2.4%)		NS
No need to reduce BZDS if there are no side effects	58 (6.0%)	30 (7.7%)	28 (4.9%)		NS
I do not know	27 (2.8%)	22 (5.6%)	5 (0.9%)		NP > P
Others	64 (6.7%)	19 (4.9%)	45 (7.9%)		NS
Non-response	4 (0.4%)	0 (0%)	4 (0.7%)		NS
Methods used to discontinue BZDs					
Gradual tapering	742 (77.1%)	233 (59.7%)	509 (89.0%)	<0.001 *	
Abrupt discontinuation	19 (2.0%)	14 (3.6%)	5 (0.9%)	0.003 *	
Switching to other psychotropic drugs	398 (41.4%)	96 (24.6%)	302 (52.8%)	<0.001 *	
Psychosocial therapy	140 (14.6%)	27 (6.9%)	113 (19.8%)	<0.001 *	
Patient materials and pamphlets	41 (4.3%)	12 (3.1%)	29 (5.1%)	0.133	
Difficulties regarding discontinuing of BZDs	904 (94.0%)	370 (94.9%)	534 (93.4%)	0.332	
Reasons for difficulties in discontinuing of BZDs					
I do not know how to discontinue	58 (6.0%)	49 (12.6%)	9 (1.6%)	<0.001 *	
I do not know when to discontinue	64 (6.7%)	50 (12.8%)	14 (2.4%)	<0.001*	
I do not know how stable the patient would be before we could discontinue	121 (12.6%)	96 (24.6%)	25 (4.4%)	<0.001 *	
Attempted to discontinue but could not because of recurrence/relapse of symptoms	537 (55.8%)	185 (47.4%)	352 (61.5%)	<0.001 *	
Attempted to discontinue but could not because of withdrawal symptoms	277 (28.8%)	65 (16.7%)	212 (37.1%)	<0.001 *	
Attempted to discontinue but could not because of the patient’s resistance	755 (78.5%)	306 (78.5%)	449 (78.5%)	0.990	

Note: *p*-values with significant results (*p* < 0.05) are indicated with an asterisk. Abbreviations: BZDs, benzodiazepine receptor agonists; NS, not significant; NP, non-psychiatrist; P, psychiatrist.

### 3.3. Factors Associated with Difficulties in Discontinuing BZDs

Table 2 shows the logistic regression analysis results examining the factors associated with difficulties in discontinuing BZDs. The difficulties in discontinuing BZDs did not differ according to the age group or specialty of the participants. In comparison with physicians who started BZD reduction immediately after symptom improvement, those who started BZD reduction 12 months or more after symptom improvement reported less difficulty in reducing BZDs (odds ratio (OR),0.203; 95% confidence interval (CI), 0.051–0.799; *p* < 0.001). Physicians who used the abrupt discontinuation method and psychosocial therapy had less difficulty in BZD discontinuation than those who did not use them (OR, 0.124; 95% CI, 0.039–0.397; *p* < 0.001; OR, 0.438, 95% CI: 0.204–0.942, *p* = 0.035, respectively). However, the difficulty in BZD discontinuation did not differ between physicians who used other BZD discontinuation methods and those who did not.

## 4. Discussion

To the best of our knowledge, this is the first study to examine differences in strategies for any difficulties with BZD discontinuation between psychiatrists and non-psychiatrists, as well as the factors related to difficulties in BZD discontinuation. This study showed that although most physicians had difficulty discontinuing BZDs regardless of specialty, the reasons for these difficulties differed between non-psychiatrists and psychiatrists. Non-psychiatrists had more difficulty with BZD discontinuation strategies, whereas psychiatrists had more difficulty with symptom recurrence/relapse and withdrawal symptoms because of BZD reduction. This study also showed that psychosocial therapy during the reduction in BZD usage was associated with lower difficulty in BZD discontinuation.

In terms of the differences between psychiatrists and non-psychiatrists, this study showed that non-psychiatrists experienced more difficulty with the strategies for BZD discontinuation, which may be related to their lower adoption of methods considered effective for BZD discontinuation, such as gradual tapering [22,23,24], psychosocial therapy [25,26,27], and switching [28,29]. However, psychiatrists reported more difficulty with symptom relapse/recurrence and withdrawal symptoms when reducing BZDs. Patients who received BZDs prescriptions from psychiatrists might have experienced more severe anxiety and insomnia, causing more frequent worsening of psychiatric symptoms with BZD reduction and greater psychiatrist difficulty. This study also showed that psychiatrists were less likely to initiate drug reduction immediately and more likely to initiate drug reduction after a year or more. A previous study using a large Japanese claims database reported that the BZD prescription by psychiatrists was a factor in predicting long-term prescription of BZDs [14,15]. The cautious psychiatrist attitudes toward BZD discontinuation revealed in this study may be one reason for the results of the previous study.

In our multivariate analysis, psychosocial therapy was associated with fewer difficulties in discontinuing BZDs. Since cognitive behavioral therapy can facilitate BZD discontinuation [25,26,27], psychosocial therapy is understandably associated with fewer difficulties in BZD discontinuation. Although we cannot discuss causal relationships because this was a cross-sectional study, educating physicians about psychosocial therapy may alleviate physician difficulty in discontinuing BZDs and reduce long-term prescriptions of BZDs. Since this study did not limit psychosocial therapy to cognitive behavioral therapy, in the future, the association between individual psychosocial therapy and difficulties in reducing medication should be investigated.

Contrary to our hypothesis, abrupt discontinuation was associated with lower difficulty in BZD reduction, while switching was associated with greater difficulty. Although a causal relationship between the difficulty and method of BZD discontinuation could not be determined, since abrupt discontinuation of BZDs may induce withdrawal symptoms and rebound insomnia or anxiety, physicians who have experienced less difficulty in discontinuing BZDs are likely to stop BZDs abruptly without the fear of side effects. Switching to psychotropic medications such as valproic acid, carbamazepine, or paroxetine has been proposed as a potential strategy for BZD discontinuation. However, robust evidence for this approach is currently unavailable [28,29]. Thus, physicians who have experienced difficulties with BZD discontinuation may try discontinuing BZDs using unestablished methods, such as switching.

This study showed that physicians who started drug reduction immediately after improvement experienced more difficulty than those who started drug reduction more than one year after initiation, in contrast to our hypothesis. Since long-term BZD usage can reduce the success rate of BZD discontinuation by increasing the risk of physical dependence, later initiation of BZD reduction, such as one year after symptom improvement [24], would not lower the difficulties with BZD discontinuation by increasing the success rate of BZD discontinuation. Although not investigated in this study, physicians who start BZD reduction immediately after improvement may experience more recurrence or relapse of symptoms in their patients, resulting in stronger resistance to BZD discontinuation from patients because of fear of recurrence or relapse. Thus, physicians experience more difficulty with BZD discontinuation in such cases than physicians who start BZD reduction more than one year after symptom improvement. Notably, psychiatrists and non-psychiatrists reported patient refusal as the most common reason for difficulties with BZD discontinuation. GPs in the United Kingdom were reported to reluctantly prescribe BZDs under pressure from patients despite not believing that they were necessary [34]. BZD discontinuation may also be true for this. Given the current lack of evidence regarding the optimal time for initiating BZD reduction, shared decision-making between patients and physicians about the need to discontinue BZDs may facilitate smooth BZD discontinuation [36].

This study has several strengths. First, this is the first study to investigate factors associated with difficulties in BZD discontinuation. Second, this study focused on non-psychiatrists and psychiatrists who tend to prescribe BZD long-term, which has not been assessed in previous studies. Third, this study examined the real-world status of BZD discontinuation strategies for which insufficient evidence exists. However, several limitations should be acknowledged. First, the survey had a low response rate, especially for the JPCA and AJHA, whose members are predominantly non-psychiatrists and were surveyed via e-mail. Moreover, the differences in response rates between psychiatrists and non-psychiatrists could have affected the results of this study. The results’ generalizability is questionable because of the low response rate. Second, the number of participants in this study was small and not representative of Japanese physicians. Third, this study did not examine successful BZD discontinuation rates or BZD prescription durations, precluding assessment of the factors related to the success rate of BZD discontinuation or the duration of BZD prescriptions. Fourth, this study did not examine physicians’ attitudes toward discontinuing individual BZDs. The differences in half-lives and other characteristics of individual BZDs might have influenced the results of this study. Fifth, this study did not investigate switching to either individual psychotropic medications or the method used for switching from BZDs. Each psychotropic may have different effects on BZD discontinuation, and the switching method may also play a role in the success rate of BZD discontinuation. Sixth, strategies for discontinuing multiple BZDs in combination may differ from strategies for discontinuing treatment with a single BZD, but this study did not collect information on single or multiple BZD therapy.

## 5. Conclusions

Most physicians experienced difficulty in discontinuing BZDs. Since existing guidelines do not provide specific strategies for BZD discontinuation, physicians may apply BZD discontinuation strategies based on their skills and experience. Further research is required to develop evidence-based BZD discontinuation strategies.

## Figures and Tables

**Table 2 ijerph-19-15990-t002:** Factors associated with difficulties in discontinuing benzodiazepine receptor agonists.

Valuable	95% CI	*p*-Value
Age groups		
20 s [Ref]		0.001
30 s	2.460 (0.170–35.649)	0.509
40 s	1.450 (0.132–15.979)	0.761
50 s	0.740 (0.072–7.569)	0.800
60 s	1.343 (0.127–14.205)	0.806
70 s	0.678 (0.061–7.502)	0.751
80 s and more	0.086 (0.007–1.086)	0.058
Specialty		
Psychiatrists [Ref: Non-psychiatrists]	0.625 (0.298–1.315)	0.216
Preferable timing to start BZDs reduction after symptoms improve		
Immediately [Ref]		
Within 3 months	1.166 (0.491–2.77)	0.728
More than 3 months but less than 6 months	0.933 (0.357–2.437)	0.887
More than 6 months but less than 12 months	1.240 (0.284–5.418)	0.775
After 12 months	0.203 (0.051–0.799)	0.023 *
No need to reduce BZDS if there are no side effects	0.528 (0.173–1.611)	0.262
I do not know	0.509 (0.113–2.287)	0.378
Others	0.891 (0.249–3.194)	0.860
Non-response	0.402 (0.014–11.28)	0.592
Methods used to discontinue BZDs		
Gradual tapering [Ref: not use]	1.398 (0.668–2.924)	0.374
Abrupt discontinuation [Ref: not use]	0.124 (0.039–0.397)	<0.001 *
Switching to other psychotropic drugs [Ref: not use]	4.839 (2.228–10.511)	<0.001 *
Psychosocial therapy [Ref: not use]	0.438 (0.204–0.942)	0.035
Patient materials and pamphlets [Ref: not use]	0.928 (0.236–3.653)	0.915

Note: *p* values with significant results (<0.05) are labeled with an asterisk. Abbreviations: BZDs, benzodiazepine receptor agonists.

## Data Availability

Data sharing is applicable if the corresponding author determines that the reason is appropriate.

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
