# Peer review of "Attitudes and Difficulties Associated with Benzodiazepine Discontinuation"

_ijerph, 2022, doi:10.3390/ijerph192315990_

Round 1

Reviewer 1 Report

The study had a clear question aimed to explore various benzodiazepine discontinuation strategies and difficulties experienced by clinicians, including psychiatrists and non-psychiatrists, and identify the factors contributing to these difficulties. Management of benzodiazepine misuse and dependence can be challenging, and gradual discontinuation is often recommended to avoid withdrawal symptoms. This is an interesting paper that gives insight into benzodiazepine discontinuation strategies used by psychiatrists and other physicians and understanding of difficulties related to discontinuation. The authors report that this is the first study in this important area (a quick literature search confirms this), and I think is well-suited for the content of the journal. The manuscript was reasonably well written, and contents seem logical (although some text would benefit from re-wording). Appropriate methods were used to answer the research question and were written with sufficient detail.

The discussion and conclusions are generally an appropriate interpretation of the results, and addresses the limitations of the study, such as poor response rates, and the relationship of the study to previous research. I note the poor response rate and would therefore question the generalisability of the results. However, it should be more concise and focused with information about specific results and implications, generalisability, and practical applications of the study.

The Discussion should to be more concise and focused with information about specific results and how the results of the study fit in or add to the existing literature.

References seem to be used appropriately, cited accurately, and formatted correctly. The text is consistent with the information presented in tables and all tables and figures included in the paper are important and relevant.

Author Response

Comments and Suggestions for Authors

The study had a clear question aimed to explore various benzodiazepine discontinuation strategies and difficulties experienced by clinicians, including psychiatrists and non-psychiatrists, and identify the factors contributing to these difficulties. Management of benzodiazepine misuse and dependence can be challenging, and gradual discontinuation is often recommended to avoid withdrawal symptoms.  This is an interesting paper that gives insight into benzodiazepine discontinuation strategies used by psychiatrists and other physicians and understanding of difficulties related to discontinuation. 

The authors report that this is the first study in this important area (a quick literature search confirms this), and I think is well-suited for the content of the journal.  The manuscript was reasonably well written, and contents seem logical (although some text would benefit from re-wording).  Appropriate methods were used to answer the research question and were written with sufficient detail. 

The discussion and conclusions are generally an appropriate interpretation of the results, and addresses the limitations of the study, such as poor response rates, and the relationship of the study to previous research.  I note the poor response rate and would therefore question the generalisability of the results. However, it should be more concise and focused with information about specific results and implications, generalisability, and practical applications of the study. 

The Discussion should to be more concise and focused with information about specific results and how the results of the study fit in or add to the existing literature.

References seem to be used appropriately, cited accurately, and formatted correctly.  The text is consistent with the information presented in tables and all tables and figures included in the paper are important and relevant.

<Response>

I appreciated your good assessment of our work. According to your contributing comment, we added that the results' generalizability is questionable because of the low response rate in the limitation section. In addition, we made sure the manuscript concisely focused on information about specific results and implications, generalizability, and practical applications of the study.

(Page 8, lines 292– 293)

The results’ generalizability is questionable because of the low response rate.

(Page 7, lines 231– 244)

In terms of the differences between psychiatrists and non-psychiatrists, this study showed that non-psychiatrists experienced more difficulty with the strategies for BZD discontinuation, which may be related to their lower adoption of methods considered effective for BZD discontinuation, such as gradual tapering [22-24], psychosocial therapy [25-27], and switching [28-29]. However, psychiatrists reported more difficulty with symptom relapse/recurrence and withdrawal symptoms when reducing BZDs. Patients who received BZDs prescriptions from psychiatrists might have experienced more severe anxiety and insomnia, causing more frequent worsening of psychiatric symptoms with BZD reduction and greater psychiatrist difficulty. This study also showed that psychiatrists were less likely to initiate drug reduction immediately and more likely to initiate drug reduction after a year or more. A previous study using a large Japanese claims database reported that the BZD prescription by psychiatrists was a factor in predicting long-term prescription of BZDs [14-15]. The cautious psychiatrist attitudes toward BZD discontinuation revealed in this study may be one reason for the results of the previous study. Long-term BZD usage is known to increase the risk of physical dependence [21], and psychiatrists’ cautious attitudes toward BZD discontinuation may lead to physical dependence via long-term prescriptions, making BZD discontinuation more difficult.

(Pages 7–8, lines 245– 253)

In our multivariate analysis, psychosocial therapy was associated with fewer difficulties in discontinuing BZDs. Since cognitive behavioral therapy can facilitate BZD discontinuation [25-27], psychosocial therapy is understandably associated with fewer difficulties in BZD discontinuation. Although we cannot discuss causal relationships because this was a cross-sectional study, educating physicians about psychosocial therapy may alleviate physician difficulty in discontinuing BZDs and reduce long-term prescriptions of BZDs. In this study, 19.8% of psychiatrists and 6.9% of non-psychiatrists used psychosocial therapy for BZD discontinuation. In Japan, cognitive behavioral therapy is only covered by insurance for depression and some anxiety disorders, not for general anxiety or insomnia, and is not widely used. Therefore, psychiatrists who use psychosocial therapy for BZD discontinuation are likely to have a high level of expertise in anxiety and insomnia or be affiliated with specialized facilities. The effectiveness of psychosocial therapy in reducing BZD withdrawal difficulties after considering confounding factors needs to be investigated in more detail. Since this study did not limit psychosocial therapy to cognitive behavioral therapy, in the future, the association between individual psychosocial therapy and difficulties in reducing medication should be investigated.

(Page 8, lines 254– 264)

Contrary to our hypothesis, abrupt discontinuation was associated with lower difficulty in BZD reduction, while switching was associated with higher difficulty. Although a causal relationship between the difficulty and method of BZD discontinuation could not be determined, since abrupt discontinuation of BZDs may induce withdrawal symptoms and rebound insomnia or anxiety, physicians who have experienced less difficulty in discontinuing BZDs are likely to stop BZDs abruptly without the fear of side effects. Switching to psychotropic medications such as valproic acid, carbamazepine, or paroxetine has been proposed as a potential strategy for BZD discontinuation. However, robust evidence for this approach is currently unavailable [28,29]. Although switching may increase BZD discontinuation difficulty by enhancing the risk of BZD discontinuation failure Thus, physicians who have experienced difficulties with BZD discontinuation may try discontinuing BZDs using unestablished methods, such as switching.

<Others>

The English language and style were corrected by an English editing company.

Reviewer 2 Report

a.    Nice article on a novel concept.

b.    Abstract: Adequately written.

c.    Introduction: Well written.

d.    Methods: How the survey was carried out? (online/ offline). How sample recruitment was done? Whether any person conducted interview of the participants?

e.    Age group 20s, who are these participants? Are they interns or graduate doctors? Do they prescribe BZDs?

f.     Any information related to use of multiple benzodiazepines together?

Author Response

Comments and Suggestions for Authors

  1. Nice article on a novel concept.
  2. Abstract: Adequately written.
  3. Introduction: Well written.

<Response>

I appreciated your good assessment of our work.

  1. Methods: How the survey was carried out? (online/ offline). How sample recruitment was done? Whether any person conducted interview of the participants?

<Response>

Thank you for this valuable comment. All members of three associations - JPCA and AJHA by e-mail and JAPC by letters - were sent questionnaires. No person interviewed the participants.

(Page 2, lines 85– 89)

We sent questionnaires between October 22, 2021 and February 1, 2022 to all physicians affiliated with the Japan Primary Care Association (JPCA), the All Japan Hospital Association (AJHA), and the Japanese Association of Neuro-Psychiatric Clinics (JAPC) through the three associations; JPCA and AJHA were contacted by e-mail and JAPC by letters.

  1. Age group 20s, who are these participants? Are they interns or graduate doctors? Do they prescribe BZDs?

<Response>

Thank you for this valuable comment. In Japan, anyone who graduates from medical school and passes the national medical examination can prescribe benzodiazepines. After qualifying as a physician, almost all physicians undergo two years of initial clinical training (rotating through various departments). After completing their initial clinical training, many physicians receive specialized training in the department of their choice and can become graduate students if they wish.

(Page 4, lines 158– 159)

There were no residents or medical students among the participants.

  1. Any information related to use of multiple benzodiazepines together?

<Response>

Thank you for this valuable comment. We did not extract information on whether benzodiazepines were used as monotherapy or in combination therapy in this survey. Strategies for discontinuing BZDs in combination may differ from strategies for discontinuing BZDs monotherapy. According to your comment, we revised the manuscript as follows:

(Page 8– 9, lines 303– 305)

Sixth, strategies for discontinuing multiple BZDs in combination may differ from strategies for discontinuing treatment with a single BZD, but this study did not collect information on single or multiple BZD therapy.

<Others>

The English language and style were corrected by an English editing company.
